# Electrochemical Immunosensor for the Quantification of S100B at Clinically Relevant Levels Using a Cysteamine Modified Surface

**DOI:** 10.3390/s21061929

**Published:** 2021-03-10

**Authors:** Alexander Rodríguez, Francisco Burgos-Flórez, José D. Posada, Eliana Cervera, Valtencir Zucolotto, Homero Sanjuán, Marco Sanjuán, Pedro J. Villalba

**Affiliations:** 1Biotechnology Research Group, Universidad del Norte, Barranquilla 081007, Colombia; alexandersanjuan@uninorte.edu.co (A.R.); fjburgos@uninorte.edu.co (F.B.-F.); edcervera@uninorte.edu.co (E.C.); hsanjuan@uninorte.edu.co (H.S.); 2Rational Use of Energy and Preservation of the Environment Group (UREMA), Universidad del Norte, Barranquilla 081007, Colombia; msanjuan@uninorte.edu.co; 3School of Medicine, Stanford University, Redwood City, CA 94063, USA; jdposada@stanford.edu; 4Gnano—Nanomedicine and Nanotoxicology Group, São Carlos Institute of Physics, University of São Paulo, São Carlos 13566-590, São Paulo, Brazil; zuco@ifsc.usp.br

**Keywords:** biosensor, gold electrodes, electrochemical impedance spectroscopy, brain injuries, S-100B, biomarker

## Abstract

Neuronal damage secondary to traumatic brain injury (TBI) is a rapidly evolving condition, which requires therapeutic decisions based on the timely identification of clinical deterioration. Changes in S100B biomarker levels are associated with TBI severity and patient outcome. The S100B quantification is often difficult since standard immunoassays are time-consuming, costly, and require extensive expertise. A zero-length cross-linking approach on a cysteamine self-assembled monolayer (SAM) was performed to immobilize anti-S100B monoclonal antibodies onto both planar (AuEs) and interdigitated (AuIDEs) gold electrodes via carbonyl-bond. Surface characterization was performed by atomic force microscopy (AFM) and specular-reflectance FTIR for each functionalization step. Biosensor response was studied using the change in charge-transfer resistance (Rct) from electrochemical impedance spectroscopy (EIS) in potassium ferrocyanide, with [S100B] ranging 10–1000 pg/mL. A single-frequency analysis for capacitances was also performed in AuIDEs. Full factorial designs were applied to assess biosensor sensitivity, specificity, and limit-of-detection (LOD). Higher Rct values were found with increased S100B concentration in both platforms. LODs were 18 pg/mL(AuES) and 6 pg/mL(AuIDEs). AuIDEs provide a simpler manufacturing protocol, with reduced fabrication time and possibly costs, simpler electrochemical response analysis, and could be used for single-frequency analysis for monitoring capacitance changes related to S100B levels.

## 1. Introduction

Traumatic brain injury (TBI) is defined as a brain dysfunction produced by an external force, usually due to a blow or sudden movement to the head, commonly occurring after a fall or traffic accidents [1]. The Center for Disease Prevention and Control (CDC) considers TBI to be one of the significant causes of disability globally, producing high annual costs to the healthcare system [2]. TBI also represents at least half of the trauma-related deaths [3] and a reported mortality rate of 37% in severe TBI [4].

Delay in the clinical identification of neurological impairment during the acute phase leads to higher mortality among TBI patients [5]. This delay could often be related to the subjective and qualitative nature of the Glasgow Coma Scale (GCS), which is routinely used as a strategy to classify the TBI severity. In addition to GCS, other invasive and non-invasive neuromonitoring techniques help to establish criteria for medical decisions. The use of these techniques requires expertise and advanced medical skills, implies potentially harmful ionizing radiation, and demands high costs for healthcare systems, limiting their availability in many resource-constrained environments, as in low-middle income countries, where the clinical examination is, in many cases, the only accessible tool for neuromonitoring [6].

The limitations above have pushed the research of blood biomarkers to improve prognosis in TBI of all severities [7]. Among investigated biomarkers for TBI, the S100B protein is perhaps the most extensively studied. S100B is a calcium-binding dimeric protein (MW: 21 kDa) primarily expressed in astrocytes and found in very low levels under physiological conditions in human cerebrospinal fluid and serum/plasma. Previous studies suggest that, following a TBI, S100B is released from damaged nerve cells into the bloodstream by passing through the blood-brain barrier (BBB), which could be disrupted after head primary injury [8].

In the clinical context, the measurement of these biomarkers demands a technique that is easy to use, readily available, low-cost, and with a rapid response time, as portable and point-of-care biosensors. In contrast, the current quantification of TBI-related biomarkers is often difficult since standard immunoassays are time-consuming, require extensive expertise and instrumentation, and usually represent a higher cost [9].

Table 1 summarizes the main biosensors tested for S100B in the last 12 years. The detection ranges used in these studies vary between concentrations from pg/mL to ng/mL. This aim to different clinical uses of biosensors, since S100B has been studied as a marker of damage in the central nervous system for other illness as Alzheimer’s disease [10,11,12], stroke [13], spinal trauma [14], and sepsis-associated encephalopathy [15] so that its use is not limited to TBI.

In this sense, the clinical significance of the S100B detection range depends on the type and severity of the brain damage [16]. Various cutoff values of S100B have been proposed for identifying brain injury [17]. Thereby, a cutoff level of 100 pg/mL [18] has been used in the mild TBI to discard the presence of intracranial hemorrhages in the computed tomography (CT), and values close to 30 pg/mL have been reported as an indicator of BBB permeability even with no associated symptoms [19]. Likewise, patients with moderate to severe TBI could display higher serum/plasmatic S100B levels in the order of ng/mL that correlates with intracranial hypertension, neurological worsening, and poor response to treatment [16].

Electrochemical-based biosensors are one of the most used systems for detecting biomarkers as S100B, primarily through faradaic process monitoring the charge transfer resistance (Rct). Currently, large macro electrodes are commonly used with monolayers to fabricate biosensors. Nevertheless, various studies have indicated the use of interdigitated electrodes (IDEs) [20,21,22,23], as they potentially offer several benefits over typical electrodes, such as lower sample volumes, lower concentrations of electro-active ions to form double layers, low ohmic drop, fast establishment of steady-state, rapid reaction kinetics, increased signal to noise ratio, and easier cleaning procedures. IDEs eliminate the need for a reference electrode and provide simple means for obtaining a steady-state current response, especially compared to the three and four-electrode set-ups, reducing the need for highly-priced instrumentation. Moreover, IDEs are easier to integrate into a complete detection system to perform parallel electrochemical assays rapidly. It should be noted that IDE-based biosensors could also offer favorable faradic-to-capacitive current ratios that lead to an improved biosensor signal-to-noise ratio, higher sensitivity, and lower detection limits for enhanced detection of antibody-antigen or aptamer target binding events [24]. Lastly, recyclable IDEs enable electrochemical detection against the same electrode system before and after analyte bonding and on each step of surface modification.

In this work, we developed two strategies for detecting and quantifying S100B, a known biomarker tested for the TBI prognosis, in both standard solutions and spiked plasma samples. The biosensors manufactured in this work are based on a zero-length cross-linking approach using EDC-NHS on cysteamine self-assembled monolayer (SAM) for the effective immobilization of anti-S100B monoclonal antibody onto both gold electrodes (AuE), and gold interdigitated electrodes (AuIDEs) via carbonyl antibody bond.

This paper focuses on the improvement of the performance of an Au-Cysteamine based biosensor in a typical three-electrode system when using recyclable AuIDEs in a two-electrodes system for the detection of the S100B protein in a buffer solution and spiked human plasma samples.

## 2. Materials and Methods

### 2.1. Electrodes and Reagents

Gold (Au) deposited onto glass substrate electrodes (AuEs) were lithographically fabricated at the National Synchroton Laboratory facilities (Campinas, Brazil) and generously facilitated by the GNano research group of the São Carlos Institute of Physics (IFSC)-USP (Brazil). Thin-film Gold InterDigitated Electrode (AuIDEs) with 180 pairs of interdigitated gold electrodes (5/5 µm, electrode/gap) were obtained from Micrux Technologies, Spain.

Etilic alcohol, acetone, 1-ethyl-3-(3-dimethylaminopropyl)carbodiimide (EDC), N-hydroxysuccinimide (NHS), phosphate-buffered saline (PBS), 30% Hydrogen Peroxide, 30% Ammonium Hydroxide, and potassium ferrocyanide (III) powder <10 um, 99% (702587) were obtained from Sigma-Aldrich (St. Louis, MO, USA). Potassium hydroxide (KOH) and cysteamine >98.0% (30070) were purchased from Sigma-Aldrich (São Paulo, Brazil). Potassium chloride (P217500) ACS 99.0 a 100% was obtained from Fisher (Hampton, NH, USA). All reagents used were of analytical grade. Deionized water (MiliQ^®^, Merck-Millipore, Molsheim, France) was employed to prepare all solutions. Recombinant Anti-S100 beta antibody [EP1576Y]-Astrocyte Marker, Recombinant Human S100 beta protein (ab55570), and Recombinant Human nNOS (neuronal) protein (ab159005) were acquired from Abcam (Cambridge, MA, USA).

### 2.2. Biosensors Construction

For the AuEs functionalization (Figure 1), they were first cleaned thoroughly using serial sonicator washes with acetone, deionized (DI) water, 2% KOH in ethanol, ethanol, and again DI water. Next, a cysteamine-SAM was grown onto clean Au WE using 30 μL of 0.5 M aqueous cysteamine solution by drop-casting onto each WE, followed by four hours incubation time at room temperature (RT). AuEs were generously rinsed with DI water and gently dried using N_2_. Subsequently, a zero-length cross-linking functionalization was performed by activating the monoclonal anti-S100B in a solution of EDC (2 mM) and NHS (5 mM) in 10 mM PBS (pH 7.4) for a final antibody concentration of 20 ug/mL. Then, 50 uL of this solution was dropped on WE and incubated for four hours at RT. Following rinsing with DI water and N_2_ drying, AuEs were blocked with 15 uL 0.5% BSA in 10 mM PBS at each WE and dried at RT for 15 min. A final washing with deionized water followed by drying with N_2_ was done.

AuIDEs functionalization with anti-S100B was performed using a modified protocol. AuIDEs were first thoroughly cleaned following RCA-1 protocol by immersion in a 5:1:1 deionized water, 27% NH_4_OH, 30% H_2_O_2_ solution [31] for five minutes, and then generously rinsed with DI water and dried using N_2_. An electrochemical impedance spectroscopy (EIS) essay was performed in potassium ferrocyanide to assess the cleaning efficiency of AuIDEs working surface, which were again rinsed with DI water and dried using N_2_. For the SAM formation, ten microliters of 0.5 M cysteamine in PBS were drop-casted on the AuIDEs WE surface and left covered at RT for 45 min on a rotary machine to improve Au-cysteamine interactions. AuIDEs were then rinsed with DI water and dried using N_2_. For the anti-S100B carboxyl group activation, a solution of 0.5 M EDC-50 ug/mL anti-S100B in PBS was prepared and vortex mixed every 15 min during two hours at RT. Then, ten microliters of the EDC-anti S100B solution were dropped-casted on each AuIDEs WE and left covered at RT for 12 h on a rotary machine to perform the anti-S100B conjugation to the Au-cysteamine SAM. Electrodes were then briefly dipped in 10 mM PBS solution three times and carefully rinsed with DI water. An EIS Test was then performed with potassium ferrocyanide to characterize the electrochemical behavior of the functionalized electrode. A subsequent blocking with ten microliters of 0.5% BSA was made by drop-casting on each AuIDE WE and left covered at RT overnight on a rotary machine. Electrodes were then thoroughly rinsed with DI water, and an EIS was done in potassium ferrocyanide to set a single reference Rct baseline for each AuIDE, which was later employed for antigen biosensing on the same electrode.

Small reagent concentrations were used in this work. Thus, to avoid undesirable fixation of them to the recipient walls, low retention tips and tubes were used to dilute, aliquoting, and drop-casting antibodies and crosslinker molecules during functionalization and further S100B tests.

### 2.3. Surface Characterization

Surface characterization of functionalized Au electrodes was carried out through atomic force microscopy (Nanosurf). Chemical characterization of functionalized WEs was performed through a specular reflectance FTIR using a Nicolet i50 FT-IR (Thermo Scientific) with the wavenumber ranging from 400 to 4000 cm^−1^.

### 2.4. Electrochemical Characterization and S100B Tests Performance

Electrochemical impedance spectroscopy was used to characterize each step of the biosensor construction and quantification of the S100B protein. S100B tests were performed by drop-casting a small volume of the sample on the WE surface of AuEs (50 uL) and AuIDEs (10 uL) and drying at RT. AuEs and AuIDEs were then connected to an M204 potentiostat/galvanostat (Metrohm^®^, Herisau, Switzerland), controlled by the NOVA 2.11 software, for the electrochemical measurements (Figure 2). EIS was conducted using an Autolab^®^ FRA32 (Methrom Company, Herisau, Switzerland) module, testing in a frequency range of 0.1 to 10,000 Hz for AuEs and 1 to 10,000 Hz for AuIDEs. Variation of Rct was recorded to evaluate changes in impedance after 15 min of antigen-antibody binding. Electrochemical measurements were carried out using 10 mM K_3_[Fe(CN)_6_] in 0.2 M KCl as a support solution, 40 uL for AuEs, and 10 uL for AuIDEs. Typical semicircular behavior in the range corresponding to high frequencies associated with the electrode redox probe was observed.

For analyte detection on AuEs, S100B samples in 10 mM PBS and spiked human plasma ranging in 10 to 1000 pg/mL were tested. For AuIDEs, only S100B spiked human plasma samples ranging in 10 to 1000 pg/mL were evaluated. In all cases, WEs were rinsed with DI water after 15 min of antigen-antibody binding, covered at RT.

### 2.5. Preparation of Buffer and Spiked Human Plasma Samples

Buffer samples were prepared by serial dilutions of the stock anti-S100B in 10 mM PBS (pH 7.4) to obtain aliquots of 10, 31, 100, 316, and 1000 pg/mL, then resuspended and frozen to −20 °C in low retention tubes. Human whole blood was obtained from a venipuncture of a healthy donor with previous consent under the Universidad del Norte (Barranquilla, Colombia) ethics committee No. 167. Whole blood, collected to Gel and EDTA K2 tubes (Improvacuter^®^ 722350202), was then centrifuged for 30 min at 10,000 g to remove proteins with MW over 30 kDa. Human plasma was extracted by pipetting and aliquoted in low retention tubes. Each aliquot was spiked with corresponding amounts of the S100B protein to obtain the same concentrations in buffer samples, and, after resuspended, they were stored at −20 °C. Each buffer and spiked human plasma sample was thawed 15 min before the test and resuspended to be drop-casted on the WE.

### 2.6. Design of Experiment (DOE)

A single factor experimental design was made to assess the effect of S100B concentration on the charge-transfer resistance (Rct) obtained from the EIS of each biosensor. A pilot test was first carried out to define the natural variability of each sensor platform, from which cleaning (Appendix A) and functionalization protocols for each platform were effectively optimized.

The biomarker S100B was tested in five levels set in a logarithmic scale, using concentrations with clinical utility: 10 pg/mL (log_10_ = 1), 100 pg/mL (log_10_ = 2), and 1000 pg/mL (log_10_ = 3), and two intermediate points according to the logarithm scale: 31 pg/mL (log_10_ = 1.5) and 316 pg/mL (log_10_ = 2.5). The change in charge transfer resistance (ΔRct) was selected as response variable, and it was defined as the difference between the Rct obtained from EIS runned for S100B testing (tRct) and the basal Rct (bRct) obtained from EIS runned on anti-S100B/BSA functionalized WE.

Since AuEs employed in this work are not reusable, the bRct for AuEs is displayed as the average of the independent measurements of a set of Cys/anti-S100B/BSA functionalized AuEs. Meanwhile, the AuIDEs were directly compared with their bRct since they could be used in successive measurements without deterioration of the signal.

Sample size (*n* = 5) was determined using the confidence interval estimation method and system variance (S^2^) was estimated from the pilot test for one level of factor ([S100B] = 100 pg/mL for AuEs and [S100B] = 31 pg/mL for AuIDEs), with a type 1 error alpha equal to 0.05 (Table 2). The meaningful difference (d), i.e., the size of the clinically relevant effect to detect, was established in 1500 Ω and 1300 Ω for AuEs and AuIDES, respectively. Operating Characteristic Curves were used with an increasing number of degrees of freedom (DOF) (replicates) to obtain type II error probability until a statistical power higher or equal to 0.9 was achieved for the given sample size.

### 2.7. Statistical Analysis

Statistical analysis was done using both RStudio and Statgraphics Centurion 18. Initially, RStudio was used to assess the assumptions of normality, homoscedasticity, and independence of residuals to establish statistical validity graphically and analytically. Then, a logarithmic transformation was applied when required to achieve a normal distribution of the experimental residues (Appendix A). We applied a one-way Welch-ANOVA to check for differences between groups, followed by Games-Howell as a post-hoc test (Appendix A). Finally, the regression models for all datasets were developed using Statgraphics Centurion 18.

All statistical tests were considered significant with a *p*-value lower than 0.05. Contour plots were also created to analyze response behavior along the region of experimentation. Afterwards, a linear regression model was performed together with a lack of fit test to determine model adequacy to each response variable. Model suitability was established considering global model significance, coefficients significance, and analysis of residuals’ structure (Appendix A).

## 3. Results

We developed a simple platform for quantifying S100B, based on a zero-length cross-linking functionalization with EDC-NHS of a monoclonal antibody anti-S100B onto AuEs and AuIDEs, modified with a cysteamine SAM for the carbonyl antibody bond. The electrochemical responses to the S100B quantification exhibit similar performance when each type of electrode was tested, displaying differences in sensibility and reproducibility.

### 3.1. Surface Characterization

#### 3.1.1. Specular Reflectance FTIR Analysis

The FTIR spectra AuEs/Cys (red) and AuE/Cys/anti-S100B (blue) are exhibited in Figure 3a. The small band ~1020 cm^−1^ in AuEs/Cys is attributed to the bending –NH2 of cysteamine, and the band at 1259 cm^−1^ is due to the C=N and C–N bonds in cysteamine. The antibodies’ successful immobilization onto SAM can be seen in the FTIR spectrum of AuE/Cys/anti-S100B. A band of amide I band at ~1700 cm^−1^ can be seen, consisting principally of C=O stretching vibration and many overlapping bands that represent different structural elements such as a-helices and b-sheets, twists, and irregular structures in no specific order [32]. Antibody immobilization is also demonstrated by the amide II bands seen between 1550–1640 cm^−1^, which consists mainly in N–H bending [33].

#### 3.1.2. Atomic Force Microscopy (AFM) Topographic Characterization

For each functionalization step, atomic force microscopy (AFM) characterization was made using the Gwyddion 2.55 software for both image editing and data analysis of surface topography. The AFM images taken on (a) AuE and (b) AuE/Cys and (c) AuE/Cys/anti-S100B surfaces are shown in Figure 4. The RMS roughness (Sq) increases as the surface is modified, from 901 pm in the bare AuEs to 2.5 nm in the anti-S100B functionalized electrode. Antibody immobilization is also evidenced by an increase in the median peak height from 4.3 nm to 20.4 nm (Table 3).

### 3.2. Electrochemical Characterization

Interface properties of working electrode surfaces of AuEs and IDEs were also evaluated through EIS. Since anti-S100B and additional molecular components of the grown biofilm are not highly conductive, a steric hindrance blocking the electron transfer of K_3_[Fe(CN)_6_]^3^^−^/^2^^−^ is formed, resulting in an increment of Rct proportional to the growth of the biofilm layer by layer either in AuS as in AuIDEs as shown in Figure 5a,b, respectively. There was also an increase in Rct when the anti-S100B functionalized surface was blocked using 0.5% BSA, so this value was the one that was finally used as basal Rct for subsequent measurements of S100B. These data are consistent with the results of surface characterization by FTIR and AFM, confirming the correct immobilization of the antibody. For AuEs, the bRct was estimated at 1685.7 ± 100.8 Ω by calculating the mean of five repetitions of the experiment. For AuIDEs, individual bRct of each electrode was used as a reference for the S100B tests.

### 3.3. S100B Measurements

Data collected from EIS spectra in the presence of 10mM K_3_[Fe(CN)_6_] redox probe for the quantification of S100B (Appendix B
Table A1) were assessed, exhibiting non-homogeneity of variance for the three conditions (AuEs-PBS, AuEs-plasma, and AuIDEs-plasma). A non-normal distribution in the AuEs dataset for the response variable (ΔRct) was observed; thus, a logarithmic transformation was applied to the AuEs dataset.

For AuEs-PBS experiments, the analysis of variance (Welch-ANOVA) and the post hoc analysis shows that the difference in the signal between each concentration tested was statistically significant (*p* < 0.05). Furthermore, Figure 6b shows the EIS spectra obtained for the quantification of S100B under condition-1 (AuEs-PBS) in the 10 to 316 pg/mL range. A proportional increment was consistently observed in the logΔRct with the successive increments of the S100B concentration. For evaluating a possible future application of these biosensors in medical diagnosis, EIS measurements of S100B were also performed in condition-2, i.e., tests in spiked human plasma samples using AuEs (AuEs-plasma) as shown in Figure 6c,d. For AuEs-plasma tests, the basal signal corresponds to the Au/Cys/anti-S100B/BSA electrode without plasma addition, while negative control refers to plasma without S100B, as displayed in Figure 6d. Results were very similar to those recorded for EIS runs performed in AuEs-PBS. EIS spectra for S100B measurements in condition-3 (AuIDEs-plasma) are displayed in Figure 7 together with the boxplot of ΔRct for the tested range of detection, also exhibiting the increasing expected behavior of ΔRct as the [S100B] increases.

Biosensors reproducibility was tested by performing five independent measurements of [S100B] = 100 pg/mL under each condition. The response of electrodes (ΔRct) was consistent and showed a relative standard deviation (RSD) of 12.6% for AuEs-PBS, 11.4% for AuEs-plasma, and 23.32% for AuIDEs-plasma, indicating good reproducibility of the S100B detection in all cases (see Appendix A).

#### 3.3.1. Curve of Calibration

A linear increment of ΔRct was observed in the range of 10 to 316 pg/mL. Higher concentrations (1000 pg/mL) showed a striking increment of ΔRct (Appendix A), which led to a marked increase in the calibration curve slope.

The response of the biosensor (y = ΔRct) to the the concentration of S100B (x = Log[S100B] (pg/mL)) for condition-1 and condition-2, in the linear detection range from 10 to 316 pg/mL is modeled by the regression equation y = 2158.48 + 3102.75 * × (*n* = 5) for AuEs-PBS tests and y = 1947.55 + 7917.07 * × (*n* = 5) for AuEs-plasma, respectively (Figure 8a,b). Each point on the calibration curve represents the average of five independent measurements and the error bar represents the standard error of the mean. The response in AuIDEs-plasma in the linear detection range from 10 to 316 pg/mL, not including 1000 pg/mL due to a nonlinear behavior, is modeled by the regression equation y = 1593.48 + 49.1927 * × (*n* = 5), where x is the [S100B] in real scale (Figure 8c).

#### 3.3.2. Limit of Detection

The limit of detection (LOD) was calculated in 18 pg/mL for AuEs-plasma and 6 pg/mL for IDEs-plasma conditions, respectively. The LOD was determined by Equation (1), where SD is the average standard deviation for each specific measurement, and m is the calibration sensitivity, determined by the slope of the calibration curve.
LOD = (3.3 × SD)/m(1)

Likewise, the results from EIS experiments comparing the response of the AuEs at the lower concentration sample in plasma (P1) against blank plasma (P0) are shown in Figure 9, exhibiting a statistically significant difference (*p* < 0.01) for the two types of electrodes.

#### 3.3.3. Specificity and Nonspecific Bindings

Figure 9a,b exhibits adequate specificity of the biosensors when the EIS is performed using the anti-S100B functionalized AuEs and AuIDEs against a different analyte. For this experiment, the nitric oxide synthase (nNOS) enzyme, another known biomarker for brain injury, was selected as the test analyte, spiked to the plasma samples to obtain a concentration of 1000 pg/mL. No significant differences between the nNOS tests and the basal signal were found (*p* = 0.86 for AuEs).

To evaluate the nonspecific bindings of the analyte in plasma samples to the BSA blocked surface, we used a newly prepared Au/Cys/BSA electrode against the higher concentration of S100B previously studied in this work. No significant statistical differences were found between the Rct in BSA-blocked surface and the Rct after the addition of S100B 1000 pg/mL using AuEs (*p* = 0.59). A similar result is observed for AuIDEs (*p* = 0.98).

#### 3.3.4. Single-Frequency Analysis (SFA)

Considering the semicircular behavior observed in the Nyquist plots of EIS experiments for AuIDEs-plasma, a single frequency analysis was developed to facilitate the monitoring of the biosensor response (Appendix A). The capacitance at each frequency was determined by Equation (2), which is given by:(2)$$C=\frac{1}{2\pi \;f\;Z^{\prime \prime }}$$
where *Z*” corresponds to the value of the imaginary portion of the impedance (measured in Ohms) obtained from the EIS at frequency f for each experimental run. The change in capacitance vs. base value was defined as:(3)$$CI\left(\%\right)=\frac{\left|C_{b}-C_{i}\right|}{C_{b}}100$$
where $C_{b}$ is the base capacitance obtained after BSA immobilization on the electrode surface, and $C_{i}$ is the capacitance obtained for each measurement at specified S100B concentrations. Figure 10 shows the percentage change of capacitance with the defined S100B concentration using specific frequencies between 1 and 10,000 Hz in a logarithmic increment. At f= 31.6 Hz, we see a continuous increase of capacitance as S100B concentrations increases.

A One-way ANOVA for *f* = 31.6 Hz was performed to check significant differences between capacitance change means for S100B concentrations between zero (blank plasma) and 1000 pg/mL. The assumptions of normality, homoscedasticity, and independence of residuals were assessed before the analysis of variance (Appendix A). Significant differences in capacitance change at *f* = 31.6 Hz were found between each S100B concentration (Appendix A). Thus, capacitance change at a single frequency could also be used as a measurement for quantifying S100B in AuIDEs. Data of single-frequency analysis (SFA) reproducibility are available on Appendix A.

## 4. Discussion

Aiming to a more straightforward method to fabricate and perform accurate quantifications of TBI-associated blood markers with an Au/Cys/anti-S100B biosensor, this work had compared a simple and scalable chemistry for surface modification in two different electrodes morphology. A simple platform for the immobilization of the monoclonal antibody via carbonyl bond was successfully developed following the EDC-NHS zero-length cross-linking approach onto a cysteamine self-assembled monolayer (SAM) grown on gold electrodes through the formation of stable Au-S bonds.

In general, an acceptable global performance was obtained for both AuEs and AuIDEs constructed S100B biosensors in terms of stability, specificity, and reproducibility. As expected, given the higher steric hindrance due to the anti-S100B and S100B protein interaction as S100B concentration increases, and electrostatic repulsive forces between the S100B and negatively charged redox species in buffer and support solution (PBS pH 7.4 and 10 mM K_3_[Fe(CN)_6_] in 0.2 M KCl) [28], a proportional increment was consistently observed in the Rct to the successive increments in S100B concentration for the two platforms tested. Additionally, we report a high specificity of our biosensors, demonstrated by a non-significant change in the signal when a different analyte is tested, most likely associated with the use of monoclonal antibodies.

Regarding the observed increment in the magnitude of signals obtained for AuEs-PBS and AuEs-plasma experiments, it could be associated to the adsorption over some BSA-free spaces in the electrode surface by small plasma proteins and other components of plasma, mainly albumine. Considering that plasma has about ten times higher albumin concentration (3.4–5.4%) than the concentration of BSA we use for blocking (0.5%), overall Rct is expected to increase with plasma when compared to PBS.

Despite having achieved the analyte’s measurement in a clinically relevant range with both electrodes’ configuration, the use of AuIDEs offers various advantages to facilitate further industrial development and commercialization of the biosensors. First, a simpler cleaning protocol (RCA-1) before functionalization is feasible for AuIDEs, avoiding expensive reagents and considerably reducing the time required. Secondly, AuIDEs possess a smaller planar detection area, which allows smaller cell volumes, thus reducing the quantity of antibody needed for biosensor functionalization without negative impact over the biosensor performance. In third place, the basal signal in AuEs must be previously estimated using the average Rct of a set of replicated tests. In contrast, for the AuIDEs, the baseline can be set for each electrode just before the analyte detection. This fact can increase the assays’ accuracy and reduce the variance and the global error with the AuIDEs approach. Note that for AuIDEs experiments, the calibration model does not require the transformation to a logarithmic scale of the value of [S100B] to observe a linear response as it does for AuEs.

Higher sensitivity and lower LOD are remarkable advantages of AuIDEs over AuEs based biosensors. An additional advantage of AuIDEs electrodes for the analyte detection lies in the fact that no reference electrode is required for the impedance measurements, making the further development of a point-of-care system easier, with less operational amplifiers as well as a simpler equivalent circuit model (Figure 11). The consistent semicircular behavior of the Nyquist diagram in AuIDEs, instead of the typical slope cause by the Warburg resistive component at low frequencies when using a three electrodes system as in AuEs, also allows a more straightforward analysis through semicircle fit to estimate the Rct, or even to perform single frequency analysis (SFA) either for the real component of impedance or for the double-layer capacitance (Cdl), as was described previously.

While EIS usually takes between two and four minutes, SFA provides a faster detection, as only one frequency point is used to find surface capacitance change. This attribute could be employed for performing non-faradaic electrochemical measurements, where sample evaporation is usually present with microfluidics

Considering that our experiments were performed in spiked plasma samples from a single donor, a lack of information about how different plasma components affect the biosensor function is evident. Hence, it is necessary to develop further studies using samples from multiple individuals, both with and without other concomitant pathologies, to evaluate the reproducibility of the high specificity shown in our results. Further studies are also needed to identify nuisance factors affecting the measurement, such as temperature, humidity, electrical noise, and tests in real settings beyond the laboratory. Likewise, more detailed studies are now underway to implement non-faradaic measurements using the AuIDEs, since the use of a redox solution could represent a limit for scaling to a commercial scale.

Even though many other studies report a wider range of detection and lower LODs (see Table 1), our work demonstrates an effective detection of S100B in a clinically relevant range in TBI patients. Those above, considering that plasmatic concentrations higher than a cutoff of 100 pg/mL [18] rules out the presence of bleeding in the CT and levels surrounding the 30 pg/mL, are associated with BBB disruption [19]. Furthermore, many of the sensors found in the literature exhibit limitations for a fast analysis due to the requirement of sandwich-immunoassays and fluorescent labels. Label-free and simple functionalization chemistry are valuable features of our biosensor, which provides a promising alternative for the fast analysis of biomarkers, even for very small volumes of samples [14].

To the best of our knowledge, this is the first S100B biosensor using a cysteamine-SAM based immobilization of the monoclonal anti-S100B antibody, which implies the use of a simple—yet effective—surface chemistry functionalization that could be used as a framework for the development of the commercial biosensor pipeline.

## 5. Conclusions

In this work, we developed a sensitive and specific biosensor for the quantification of the brain injury biomarker S100B in clinically relevant levels, based on a simple chemistry of functionalization. Our results show an overall adequate performance and reproducibility for both AuEs and AuIDEs based biosensors. To the best of our knowledge, this is the first S100B biosensor using a cysteamine-SAM based immobilization of the monoclonal anti-S100B antibody.

AuIDEs-S100B biosensor offers a simpler manufacturing protocol, good reproducibility, short response time, reduced fabrication times, and possibly low costs. The strategy presented here could be a valuable framework for the design and fabrication of new immunosensors to detect other biomarkers of clinical interest.

## Figures and Tables

**Figure 1 sensors-21-01929-f001:**
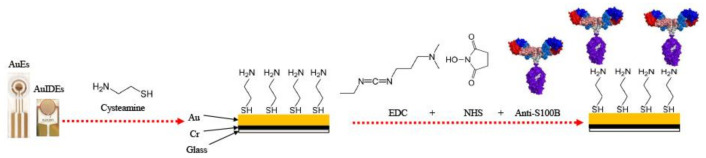
Graphical abstract of the covalent immobilization of anti-S100B onto both gold electrodes (AuEs) and gold interdigitated electrodes (AuIDEs) platforms using a cysteamine/EDC-NHS approach.

**Figure 2 sensors-21-01929-f002:**
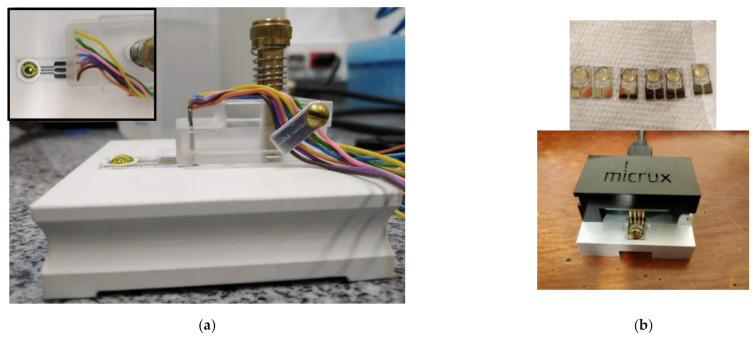
(**a**) Set-up of AuEs connection to M204 potentiostat/galvanostat for S100B measurement. The 10 mM K_3_[Fe(CN)_6_] in 0.2 M KCl solution was added just before the measurement is observed. The inserted picture shows the superior view, and the circular working area was outlined with self-adhesive paper. (**b**) Picture of AuIDEs during one of the functionalization steps (above), the circular working area is defined by SU-8 resin and the set-up connection to M204 potenti-ostat/galvanostat using the Micrux technologies Drop-Cell connector (below). As on AuEs, the redox solution is added just before measurements.

**Figure 3 sensors-21-01929-f003:**
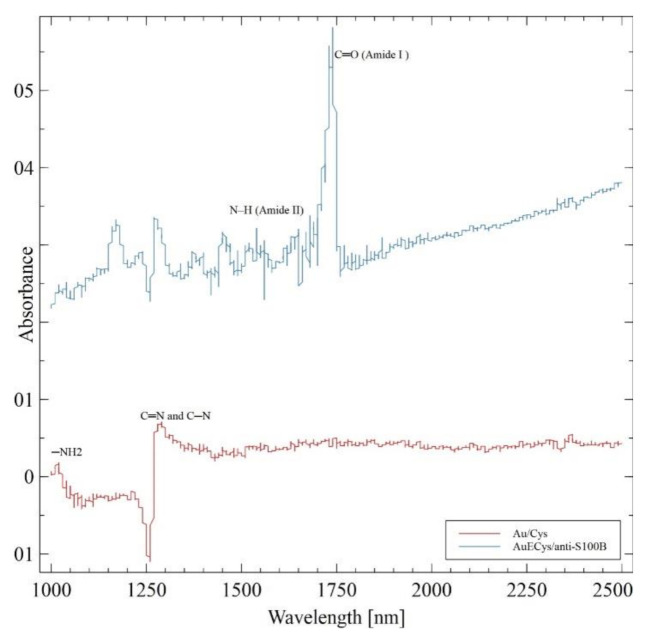
Specular reflectance FTIR spectra for AuEs/Cys (red) and AuE/Cys/anti-S100B (blue). Bands at ca. 1020 cm^−1^: –NH2 bend in cysteamine; 1259 cm^−1^: C=N and C–N bonds in cysteamine; 1550–1640 cm^−1^: amide II, N–H bending; ~1700 cm^−1^: amide I band, principally of C=O stretching. Both spectra were normalized, and the background was subtracted before comparisons.landmark.

**Figure 4 sensors-21-01929-f004:**
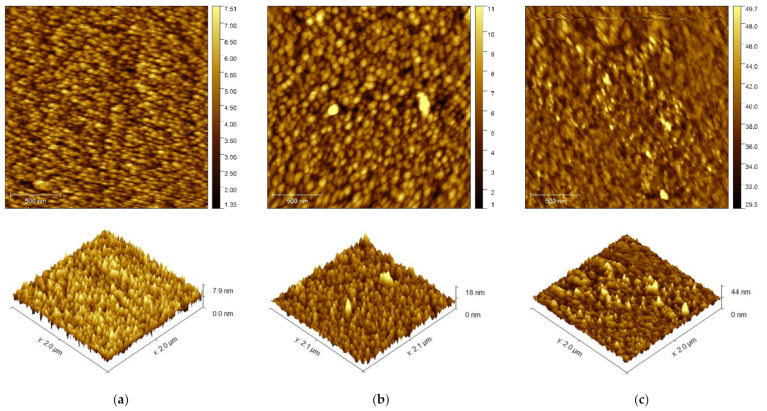
2D and 3D AFM topography (**a**) AuE and (**b**) AuE/Cys and (**c**) AuE/Cys/anti-S100B.

**Figure 5 sensors-21-01929-f005:**
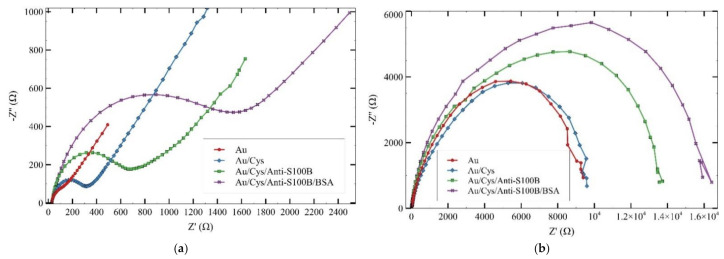
Nyquist plots for the consecutive steps of functionalization in (**a**) AuEs and (**b**) AuIDES. An increment of charge-transfer resistance (Rct) proportional to the growth of the biofilm is observed in both platforms.

**Figure 6 sensors-21-01929-f006:**
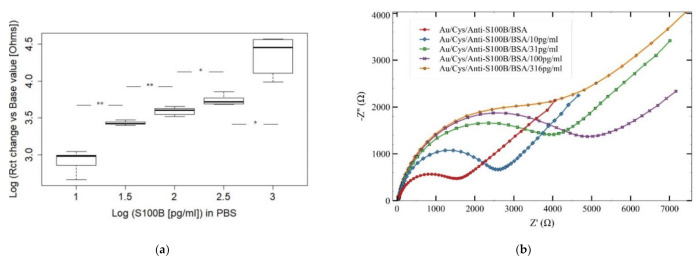
Data in brief for the S100B tests. (**a**) boxplot of ΔRct values for measurements of [S100B] in PBS (pH 7.4) at the 10–1000 pg/mL linear range of detection. (**b**) Nyquist plots for AuEs-PBS tests in the same range. The results for AuEs-plasma are presented in (**c**, **d**). Significance (Games-Howell test): *p* < 0.01 (*); *p* < 0.001 (**); *p* = 0.000 (***); *x*-axis in (a) = Log[S100B] (pg/mL), *y*-axis in (**a**) = LogΔRc(Ω); *x*-axis in (**b**) = Z’(Ω), *y*-axis in (**b**) = −Z’’(Ω).

**Figure 7 sensors-21-01929-f007:**
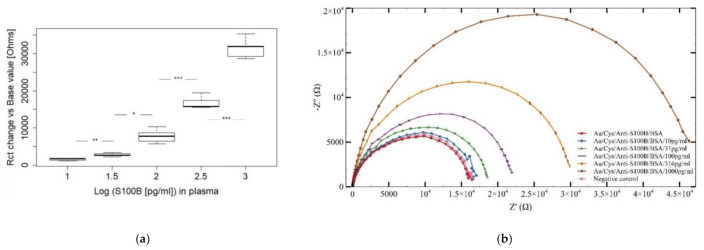
(**a**) Boxplot and (**b**) Nyquist plots of AuIDEs experimental results in spiked human plasma samples for the quantification of S100B in the 10–1000 pg/mL range. Significance (Games-Howell test): *p* < 0.01 (*); *p* < 0.001 (**); *p* = 0.000 (***); *x*-axis in (**a**) = Log[S100B] (pg/mL), *y*-axis in (**a**) = ΔRc(Ω); *x*-axis in (**b**) = Z’(Ω), *y*-axis in (**b**) = - Z’’ (Ω).

**Figure 8 sensors-21-01929-f008:**
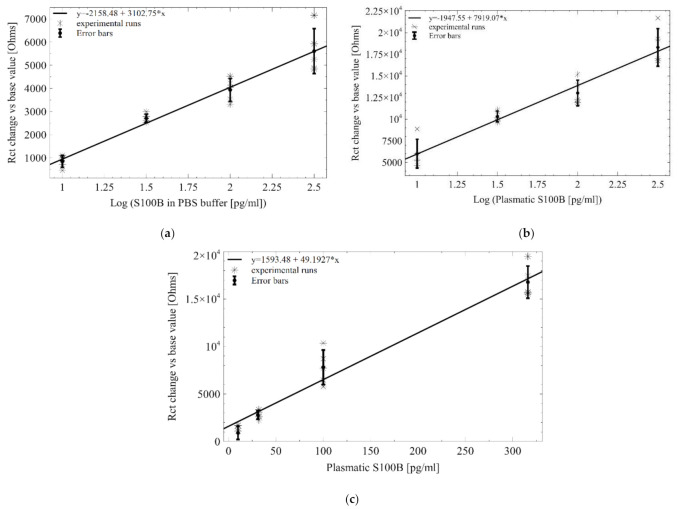
Curves of calibration for S100B tests (**a**) in 10 mMPBS (pH 7.4) samples using AuEs, (**b**) in spiked human plasma samples using AuEs and (**c**) in spiked human plasma using Au IDEs. y = ΔRct; x = [S100B] (pg/mL).

**Figure 9 sensors-21-01929-f009:**
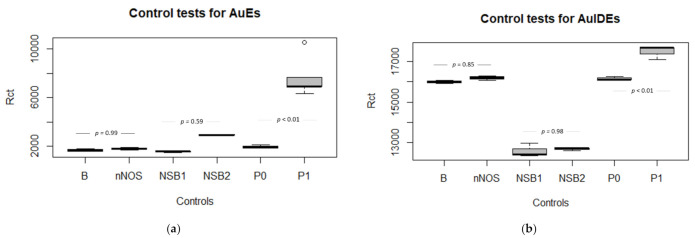
(**a**) Control tests comparing the Rct in AuEs biosensors for plasma samples; B = basal Rct in the anti-S100B functionalized AuEs; nNOS = Rct for the nNOS (1000 pg/mL) tests using the anti-S100B functionalized AuEs, indicating good specificity; P0 = plasma blank, P1 = the lowest tested S100B concentration (10 pg/mL). NSB1 = the basal Rct signal in the Au/Cys/BSA electrode. NSB2 = Rct after the addition of S100B at 1000 pg/mL to the Au/Cys/BSA electrodes, indicating the nonspecific bindings to the blocked platform in the absence of anti-S100B. (**b**) These findings are also reproducible in the AuIDEs for specificity and nonspecific bindings.

**Figure 10 sensors-21-01929-f010:**
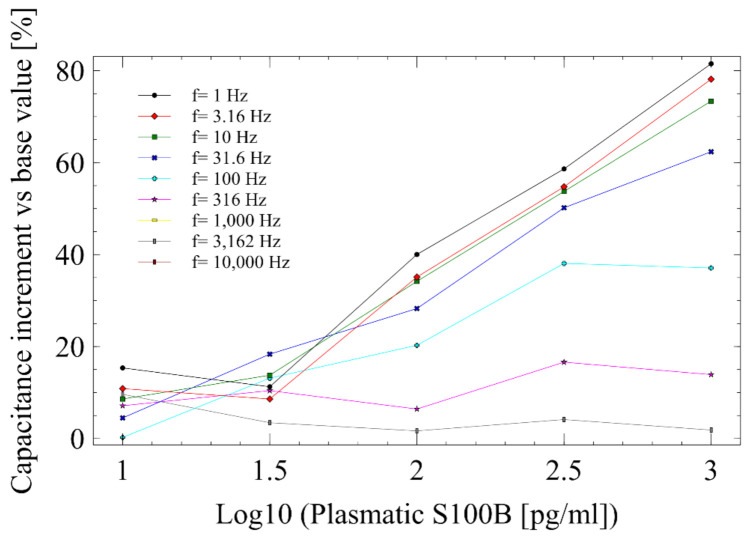
Capacitance analysis of the AuIDEs-S100B biosensors, using a single frequency analysis approach.

**Figure 11 sensors-21-01929-f011:**
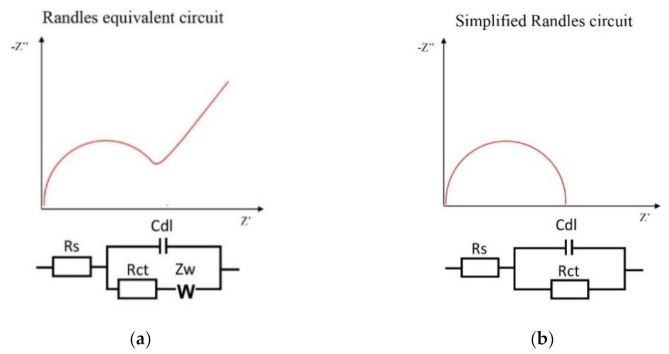
Schematic representation of the equivalents circuits for the Nyquist plots obtained from EIS performed using (**a**) AuEs biosensors, corresponding to a typical Randles circuit and using (**b**) AuIDEs in which a simplified Randles circuit fit the plot. Rs, solution resistance; Rct, charge-transfer resistance; Cdl, double-layer capacitance; Zw, Warburg impedance.

**Table 1 sensors-21-01929-t001:** Summary of S100B biosensors developed in the last 12 years and their analytical performances.

Author (year)	Chemistry of Functionalization	Detection Method	Range of Detection	LOD
Kim et al., (2015) [9]	MB-Ab-S100B-Ab-QD	Sandwiched immunocomplex optical detection	0.01–30 ng/mL	10 pg/mL
Mikuła et al., (2014) [11]	Au/(NAC-DPTA)-Cu(II)/His6-RAGE VC1 or C2 domains	Electrochemical	370–7.4 ng/mL	193 pg/mL * (buffer) 240 pg/mL * (plasma)
Kurzątkowska et al., (2016) [10]	Au/NAC/DPM–Cu(II)/His-tagged RAGE	Electrochemical	333–7.4 ng/mL *	963 pg/mL * (buffer) and 333 pg/mL *(plasma)
	Au/MBT/DPM–Cu(II)/His-tagged RAGE	Electrochemical	1–7.4 ng/mL *	1.8 ng/mL * (buffer) and 1 ng/mL * (plasma)
Tabrizi et al., (2019) [12]	anti-S100B/rGO-Au/ITO	Sandwich-type photoelectrochemical immunoassay	0.25–1000 pg/mL	0.15 pg/mL
Harpaz et al., (2019) [13]	Au/MUA-EDC/NHS-anti-S100B	SPR biosensor	0.25–10 ng/mL	0.75 ng/mL (water) and 0.136 ng/mL (plasma)
Khetani et al., (2017) [14]	GSPE/4-NBD/GA/anti-S100B	Electrochemical immunosensor	1–10 ng/mL	1 pg/mL
Lee et al., (2009) [25]	CNT/PBSE/anti-S100B	Electrical immunosensor	1–100 ng/mL	Not reported
Liu et al., (2013) [26]	PEI-PMMA/anti-S100B	Microchip-based electrochemical immunosensor	0.1–100 pg/mL	0.1 pg/mL
Cardinell et al., (2019) [27]	GDE/16-MHDA/anti-S100B	Electrochemical	1–1000 pg/mL	2–5 pg/mL (purified solution) and 14–67 pg/mL (spiked plasma)
Y.-C. Kuo et al., (2018) [28]	IDZB/Cys/GA/anti-S100B	Electrochemical	10–10 μg/mL *	10 ng/mL *
Mathew et al., (2018) [29]	SPEs/CNT-nafion-GA/Ab-S100B-Ab	Electrochemical immunoassay —FEED	10–10 ng/mL	10 fg/mL
Hassanai, et al., (2020) [30]	Au-coated magnetic NPs/thiol-ended anti-S100B fragments	Electrochemical	3.7–37 ng/mL *	3.7 ng/mL *

* Converted from the originally reported concentration in pM. Abbreviations: rGO-Au: Green reduced graphene oxide and decorated with gold nanoparticles; CNT/PBSE: Carbon nanotubes/1-pyrenebutanoic acid succinimidyl ester; PEI-PMMA: poly(ethyleneimine) modified poly(methyl methacrylate); (NAC/DPTA)–Cu(II): N-Acetylcysteamine/Thiol Derivative of Pentetic Acid-Cu(II) Monolayer; GDE/16-MHDA: Gold Disk Electrodes/16-mercaptohexadecanoic acid; MB-Ab-S100B-Ab-QD: antibody-conjugated magnetic beads (for capture), and antibody-conjugated quantum dots (for optical detection); GSPE/4-NBD/GA: Graphene screen-printed electrode/4-nitrobenzenediazonium salt/glutaraldehyde; SPR: Surface Plasmon Resonance; DPM: dipyrromethene; MBT:4-mercaptobutanol; IDZB/Cys/GA: interdigitate-zigzag biochip/cysteamine/glutaraldehyde; SPEs: screen printed electrodes; FEED: Field effect enzymatic detection.

**Table 2 sensors-21-01929-t002:** Sample size calculation for AuEs and AuIDEs electrodes.

**Gold Electrodes (AuEs) ^1^**						
***n***	**fi2**	**fi**	**OC Curve (v1)**	**DOF (v2)**	**Beta**	**Power**
3	3.270211	1.808372	4	10	0.33	0.67
4	4.360281	2.088129	4	15	0.175	0.825
5	5.450351	2.334599	4	20	0.061	0.939
**Interdigitated Gold Electrodes (AuIDEs) ^2^**						
***n***	**fi2**	**fi**	**OC Curve (v1)**	**DOF (v2)**	**Beta**	**Power**
3	2.666285	1.632876	4	10	0.36	0.64
4	3.555047	1.885483	4	15	0.175	0.82
5	4.443809	2.108034	4	20	0.061	0.935

^1^ Estimated system variance (S^2^) at 245,194.945; ^2^ estimated S^2^ at 190,152.2.

**Table 3 sensors-21-01929-t003:** AFM parameters for the topographic characterization of each functionalization step of AuE/Cys/anti-S100B biosensors.

Surface	Sq(nm) ^1^	Sa(nm) ^2^	Ssk ^3^	Median Peak Height (nm)
Au	9.012 × 10^−^^4^	7.240 × 10^−^^4^	1.577 × 10^−^^2^	4.313
Au-Cys	1.639	1.276	0.578	5.875
Au-Cys-anti-S100B	2.517	1.876	1.119	20.401

^1^ RMS roughness; ^2^ Mean roughness; ^3^ skew.

## Data Availability

Data supporting reported results can be found in http://doi.org/10.5281/zenodo.4445966 for publicly archived datasets analyzed in this work. Also, the R code is available in https://github.com/arodriguez83/S100B_biosensor.git.

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
