# Peer review of "Electrochemical Immunosensor for the Quantification of S100B at Clinically Relevant Levels Using a Cysteamine Modified Surface"

_sensors, 2021, doi:10.3390/s21061929_

Round 1
Reviewer 1 Report
In this paper, Rodriguez et al. developed an electrochemical immunosensor for the quantification of S100B at clinically relevant levels using a cysteamine modified surface. It is very interesting, I should accept with major revision and I have some considerations:
- In my opinion, the introduction section is too wide, and the table is enough to show the state of the art. Perhaps it is better to reduce the length of the introduction summing it all up.
- Figure 1 is too small and not easily readable.
- The authors perform the drop-casting of small volumes of the sample on the WE surface of AuEs (50uL) and AuIDEs (10uL) and drying at RT. Are the authors sure that the drop placed is found only on the working electrode and does not transfer out? Can the authors to show data of reproducibility?
- Is the Rct of the electrodes treated with EDC/NHS the same for all electrodes? Usually, when you use this type of electrodes, it is not the same and it is preferable to consider the relative values and not the absolute values of the Rct. This is why depending on the start value; you can obtain a sort of amplification for the next steps and it results impossible to consider absolute values.
- The authors to avoid several issues due to non-specific interactions, they used the blocking with BSA. In fact, in figure 5a (not so clear because not easily readable) the increase in Rct due to BSA is huge. Why the authors did not improve the functionalization step with antibodies increasing, possibly, the sensitivity and the LOD?
- The figures 6 and 7 are not easily readable.
- I should suggest to review all figures because they are not clear, the fonts are too small.
Author Response
RESPONSE TO REVIEWERS
Article Title: Electrochemical immunosensor for the quantification of S100B at clinically relevant levels using a cysteamine modified surface
Manuscript Number: sensors-1097907
We would like to thank the reviewers for their constructive comments. The manuscript has been thoroughly revised and rewritten to address the questions below. In the “Marked_manuscript” document we have used tracked changes in Microsoft word to highlight changes made in the document.
In this point-by-point response letter we have employed quotation marks for showing changes made in the manuscript for each of the reviewer’s comments and indicating the specific line of the highlighted manuscript, where the new introduced text is underlined.
Reviewer 1
In this paper, Rodriguez et al. developed an electrochemical immunosensor for the quantification of S100B at clinically relevant levels using a cysteamine modified surface. It is very interesting, I should accept with major revision and I have some considerations:
1. In my opinion, the introduction section is too wide, and the table is enough to show the state of the art. Perhaps it is better to reduce the length of the introduction summing it all up.
We have focused the introduction, editing some sentences as follows:
Lines 39: “...producing high”
Lines 40-41: The following sentence was eliminated: “of more than 75 billion dollars in patient treatment, rehabilitation, and loss of productivity”
Lines 42-43: Changed by “...related deaths [3] and a reported mortality rate of 37% in severe TBI [4].”
Lines 47-48: “...other invasive and non-invasive ...”
Lines 49-51: The following sentence was eliminated: “Among these are the imaging techniques for protocolized use such as CT, MRI, and used in more severe cases the intracranial pressure (ICP) monitoring, transcranial döppler, and venous saturation of O2 in the jugular bulb (SjO2)”
Lines 56-57: The following sentence was eliminated: “and other biofluids”
Lines 64 - 66: Changed by “In the clinical context, the measurement of these biomarkers demands a technique that is easy to use, readily available, low-cost, and with a rapid response time, as portable and point-of-care biosensors. In contrast, the current quantification of TBI-...”
Lines 69-72: The following sentence was eliminated “For this reason, the development of highly sensitive biosensors is required to quantify these biomarkers. On the other hand, the clinical context in which the measurement of these biomarkers is needed demands a measurement technique that is easy to use, readily available, low-cost, and with a rapid response time, as portable and point-of-care biosensors”.
Lines 83-84: Added to clarify abbreviation “computed tomography (CT)”
Lines 117-118: changed by “...the TBI prognosis, in both...”
Lines 122-123: The following sentence was eliminated “Different concentrations of S100B were tested, and the results are displayed as the average of the independent measurements”.
2. Figure 1 is too small and not easily readable.
We have increased the size of Figure 1 by 50% to make it readable..
3. The authors perform the drop-casting of small volumes of the sample on the WE surface of AuEs (50uL) and AuIDEs (10uL) and drying at RT. Are the authors sure that the drop placed is found only on the working electrode and does not transfer out? Can the authors to show data of reproducibility?
AuEs have a defined WE surface which is surrounded by a blank circular adhesive to maintain a confined working area for all electrodes. Similarly, AuIDEs are manufactured with a SU-8 film which accurately defines the working area. Hence, when we drop cast small volumes of the sample on the working area, we consider no sample is transferred to places differentes from the working area since no leakage is perceived during the measurement. Data about reproducibility was added to the supplementary material of the manuscript as Table S10, which provides information about the variance of each treatment level in the experimental design.
4. Is the Rct of the electrodes treated with EDC/NHS the same for all electrodes? Usually, when you use this type of electrodes, it is not the same and it is preferable to consider the relative values and not the absolute values of the Rct. This is why depending on the start value; you can obtain a sort of amplification for the next steps and it results in impossible to consider absolute values.
We employ relative values for each Rct measurement. These values are calculated relative to the Rct obtained for each electrode after functionalization is performed (after immobilization of BSA). Hence, amplification is found relative to this Rct value. Lines 236-239 describe this part of the methodology i.e. “The change in charge transfer resistance (ΔRct) was selected as response variable, and it was defined as the difference between the Rct obtained from EIS runned for S100B testing (tRct) and the basal Rct (bRct) obtained from EIS runned on anti-S100B/BSA functionalized WE.”
5. The authors to avoid several issues due to non-specific interactions, they used the blocking with BSA. In fact, in figure 5a (not so clear because not easily readable) the increase in Rct due to BSA is huge. Why the authors did not improve the functionalization step with antibodies increasing, possibly, the sensitivity and the LOD?
Since the proposed utility of our biosensor is the measurement in ranges of clinical interest, all S100B values below our LOD are considered noise, which would correspond to physiological values. Considering that one of our goals is to improve measurement capability without significantly increasing biosensor manufacturing costs, higher antibody concentration would increase the costs of the biosensor. In addition, our pilot experiments showed that the anti-S100B concentration used is acceptable to obtain a reproducible signal and it is in the range of those previously reported in the literature for other immunosensors, i.e: Lee, H. S.; Oh, J. S.; Chang, Y. W.; Park, Y. J.; Shin, J. S.; Yoo, K. H. Carbon Nanotube-Based Biosensor for Detection of Matrix Metallopeptidase-9 and S-100B. Curr. Appl. Phys. 2009, 9 (4 SUPPL.), e270–e272.
6. The figures 6 and 7 are not easily readable.
Figures 6 and 7 have been remade to make it readable. Fonts size was increased to allow better readability. Additionally, the inserts in Figure 6(b) and 6(d) were separated and showed as an independent figure in the supplementary material. Modifications in the text manuscript were made when referring to the inserts (lines 326, 328 and the Figure 6 caption).
7. I should suggest reviewing all figures because they are not clear, the fonts are too small.
All figures have been reviewed and modified to make them more readable when it was necessary, in such a way that font size was increased to allow better readability. Specifically, the following figures were modified:
Figure 1: Size was increased in 50%
Figure 3: Size was increased in 50%
Figure 5: Fonts size was increased
Figure 6: Fonts size was increased. The inserts in 6(b) and 6(d) were separated and showed as an independent figure in the supplementary material.
Figure 7: Fonts size was increased and we have also added the EIS for the negative control in AuIDEs.
Figure 8: The total size of the panel was increased to the width of the document window
Reviewer 2 Report
Review report of the manuscript titled Electrochemical immunosensor for the quantification of S100B at clinically relevant levels using a cysteamine modified surface.
The reported work is interesting, but in my opinion, different and relevant aspects have to be clarified before recommendation for publishing in the Sensors Journal.
In line 273, authors say that bare AuE spectra in exhibited in Figure 3(b). However, I couldn’t find this figure.
Related to Table A1, it is difficult to understand the information that authors try to provide. Perhaps, a reorganization of data according to their concentration, could help.
In figure 6d appears a negative control. Which is the difference with the signal for Au/Cys/Anti-S100B/BSA? It has to be clarified. Why the negative control is not shown in Figure 7 for AuIDEs?
At least a comment has to be done related to the differences in the magnitude of signals obtained for AuEs-PBS and AuES-plasma, and how the system is affected by the complexity of the matrix. If there is not non-specific adsorption, why the signals are so different for plasma samples related to buffer ones?
In figure 8b, the Y-axis are not clear.
Related to the biosensor’s reproducibility (lines 338-342), how the experiments are performed? In the discussion of the results (line 442), authors claim that AuIDEs are clearly superior in reproducibility. It is not clear whit the results provided (RSD 23,32%).
It is also not clear what part from AuIDEs are reused for successive measurements. In the RCA-1 cleaning protocol are eliminated all the functionalization steps? In case, where is the advantage in time for their construction and performance? If not, have you made any stability experiment? What about the breaking of antibody-antigen interaction? Please, clarify this aspect.
Authors also claims the advantages of performing single-frequency analysis. However, no comparison of the results is provided.
Only calibration curves with spiked plasma samples are obtained. In a biosensor design developed for clinical analysis, at least, some real clinical samples have to be evaluated. Author have to present the results of these kind of samples.
Author Response
RESPONSE TO REVIEWERS
Article Title: Electrochemical immunosensor for the quantification of S100B at clinically relevant levels using a cysteamine modified surface
Manuscript Number: sensors-1097907
We would like to thank the reviewers for their constructive comments. The manuscript has been thoroughly revised and rewritten to address the questions below. In the “Marked_manuscript” document we have used tracked changes in Microsoft word to highlight changes made in the document.
In this point-by-point response letter we have employed quotation marks for showing changes made in the manuscript for each of the reviewer’s comments and indicating the specific line of the highlighted manuscript, where the new introduced text is underlined.
Reviewer 2
Review report of the manuscript titled Electrochemical immunosensor for the quantification of S100B at clinically relevant levels using a cysteamine modified surface.
The reported work is interesting, but in my opinion, different and relevant aspects have to be clarified before recommendation for publishing in the Sensors Journal.
- In line 273, authors say that bare AuE spectra in exhibited in Figure 3(b). However, I couldn’t find this figure.
We decided to remove bare Au spectra, which was shown in panel b, since figure 3 was too large. However, the sentence that described remained in the text by mistake. Hence, following the reviewer’s comment, we have deleted the sentence.
- Related to Table A1, it is difficult to understand the information that authors try to provide. Perhaps, a reorganization of data according to their concentration, could help.
We have reorganized the information provided in Table A1 according to S100B concentrations, as suggested by the reviewer.
- In figure 6d appears a negative control. Which is the difference with the signal for Au/Cys/Anti-S100B/BSA? It has to be clarified. Why the negative control is not shown in Figure 7 for AuIDEs?
The negative control refers to plasma without S100B (also shown in Figure 8 as “P0”). Au/Cys/Anti-S100B/BSA (also referred as “B” in Figure 8) corresponds to the basal signal without adding plasma, just the redox probe. Following the reviewer’s comment, we have provided clarification between the signal for Au/Cys/Anti-S100B/BSA and negative control (line 333-335). We have also added the EIS for the negative control for AuIDEs in figure 7.
- At least a comment has to be done related to the differences in the magnitude of signals obtained for AuEs-PBS and AuES-plasma, and how the system is affected by the complexity of the matrix. If there is not non-specific adsorption, why the signals are so different for plasma samples related to buffer ones?
Plasma contains proteins, such as albumin, and other components, i.e.,glucose, clotting factors, electrolytes (Na+, Ca2+, Mg2+, HCO3−, Cl−, etc.), hormones, carbon dioxide, and oxygen. This implies more adsorpiont of molecules on the WE with plasma, since some BSA-free spaces in the electrode surface can be occupied by small plasma proteins or other components of plasma, mainly albumine. Considering that plasma has 10 times higher albumin concentration (3.4 - 5.4%) than the concentration of BSA we use for blocking (0.5%), overall Rct is expected to increase with plasma when compared to PBS. A comment related to this issue was added in lines 438-443.
- In figure 8b, the Y-axis are not clear.
We have remade Figure 8 and increased font size to provide more readability to the figure.
- Related to the biosensor’s reproducibility (lines 338-342), how the experiments are performed? In the discussion of the results (line 442), authors claim that AuIDEs are clearly superior in reproducibility. It is not clear with the results provided (RSD 23,32%).
After reviewing lines 338-342 and line 442 of the revised manuscript, we could not identify what the reviewer suggested. Higher reproducibility with AuIDEs when compared to AuEs was not declared on our manuscript.
Line 430 of the highlighted manuscript states: “an acceptable global performance was obtained for both AuEs and AuIDEs constructed S100B biosensors in terms of stability, specificity, and reproducibility”. Hence, no reference is made to higher reproducibility of AuIDE versus AuE.
In case another question regarding this inquiry arises, we would be more than willing to provide an answer to it.
- It is also not clear what part from AuIDEs are reused for successive measurements. In the RCA-1 cleaning protocol are eliminated all the functionalization steps? In case, where is the advantage in time for their construction and performance? If not, have you made any stability experiments? What about the breaking of antibody-antigen interaction? Please, clarify this aspect.
AuIDEs are reused in two ways:
- Consecutive EIS measurements, first with basal BSA value and then after sample is drop casted. This case is for obtaining relative Rct values (as explained in lines 240-243 of the highlighted manuscript).
- New experimental runs, where a previously used electrode is again cleaned with RCA-1 cleaning solution to remove all organics compounds. This part is established by performing another EIS on the bare electrode, which shows a RCT almost identical to the one found after the first RCA-1 cleaning during the first experimental run.
The advantages mainly lie in two aspects. 1. AuIDEs require a five minutes cleaning protocol with low priced chemicals before functionalization begins. 2. By allowing consecutive measurements, relative Rct values are obtained for each specific basal Rct value for each AuIDE, improving performance.
Stability experiments were done by performing three consecutive cleaning, functionalization and measurement on an unique electrode to see whether Rct changed between each experimental run.
- Authors also claim the advantages of performing single-frequency analysis. However, no comparison of the results is provided.
We have added a results comparison between EIS analysis and Single Frequency Analysis for AuIDEs in the discussion section, lines 470-473 stating: “While EIS usually takes between two and four minutes, SFA provides a faster detection, as only one frequency point is used to find surface capacitance change. This attribute could be employed for performing non faradaic electrochemical measurements, where sample evaporation is usually present with microfluidics.”
- Only calibration curves with spiked plasma samples are obtained. In a biosensor design developed for clinical analysis, at least, some real clinical samples have to be evaluated. Author have to present the results of these kind of samples.
At this moment we are not able to provide these results since we are still on a proof of concept stage in the development cycle of the biosensor according to the technology readiness levels (TRLs). Our biosensor currently complies with TRL 4, which refers to an Alpha prototype of S100B detection system tested with S100B spiked blood plasma sample in laboratory settings.
In addition, a clinical study was begun in march 2019 with TBI patients, where blood was extracted for S100B detection with the proposed biosensor. However, due to the Covid-19 pandemic, the study was stopped and has not resumed as of now. We expect to begin tests of TRL 5 (Alpha prototype of S100B detection system tested with TBI patients' blood plasma samples in relevant settings) once the current public health situation in our Country allows us.
Round 2
Reviewer 1 Report
The article is well done. I think it is fine for publication
Reviewer 2 Report
After adequate revision, in my opinion, the manuscript can be published by the Journal in the present form.